# Long-range charge carrier mobility in metal halide perovskite thin-films and single crystals via transient photo-conductivity

Jongchul Lim [1,2,4✉], Manuel Kober-Czerny [1,4], Yen-Hung Lin [1], James M. Ball[1], Nobuya Sakai[1], Elisabeth A. Duijnstee[1], Min Ji Hong[3], John G. Labram [3], Bernard Wenger [1✉] & Henry J. Snaith [1✉]

Charge carrier mobility is a fundamental property of semiconductor materials that governs many electronic device characteristics. For metal halide perovskites, a wide range of charge carrier mobilities have been reported using different techniques. Mobilities are often estimated via transient methods assuming an initial charge carrier population after pulsed photoexcitation and measurement of photoconductivity via non-contact or contact techniques. For nanosecond to millisecond transient methods, early-time recombination and exciton-to-free-carrier ratio hinder accurate determination of free-carrier population after photoexcitation. By considering both effects, we estimate long-range charge carrier mobilities over a wide range of photoexcitation densities via transient photoconductivity measurements. We determine long-range mobilities for $FA_{0.83}Cs_{0.17}Pb(I_{0.9}Br_{0.1})_3$, $(FA_{0.83}MA_{0.17})_{0.95}Cs_{0.05}Pb(I_{0.9}Br_{0.1})_3$ and $CH_3NH_3PbI_{3-x}Cl_x$ polycrystalline films in the range of 0.3 to 6.7 $cm^2\,V^{-1}\,s^{-1}$. We demonstrate how our data-processing technique can also reveal more precise mobility estimates from non-contact time-resolved microwave conductivity measurements. Importantly, our results indicate that the processing of polycrystalline films significantly affects their long-range mobility.

[1] Clarendon Laboratory, Department of Physics, University of Oxford, Parks Road, Oxford OX1 3PU, UK. [2] Graduate school of Energy Science and Technology, Chungnam National University, 99 Daehak-ro, Daejeon 34134, Republic of Korea. [3] School of Electrical Engineering and Computer Science, Oregon State University, Corvallis, OR 97331, USA. [4]These authors contributed equally: Jongchul Lim, Manuel Kober-Czerny. ✉email: jclim@cnu.ac.kr; bernard.wenger@physics.ox.ac.uk; henry.snaith@physics.ox.ac.uk

For optimising the performance of optoelectronic devices like photovoltaics (PV), light-emitting diodes (LED), photo-detectors and transistors, and in order to understand the best field of use for a material, it is important to understand the long-range optoelectronic properties of the semiconductor material, and if these properties change in different carrier density regimes[1–5]. Amongst many other factors, including light absorption properties, charge carrier lifetime and photo-luminescence quantum efficiency, the charge carrier mobility is a key material parameter for understanding charge transport in different internal carrier density regimes[4,6–11].

Many different characterisation methods have been employed in order to determine reliable mobility values based on transient, steady-state, contact and non-contact configurations[7,8,10–20]. Field-effect transistors (FETs)[7] and space-charge-limited-current (SCLC)[8,14,21] diodes probe the steady-state mobility through electrical contacts. FETs are limited in their ability to estimate bulk mobilities in perovskites because both the semiconductor-dielectric interface and ionic displacement in the channel can affect electronic charge transport and estimations of accumulated charge density. In addition, FETs only operate in a higher charge carrier density regime which is beyond the operating conditions for many semiconductor devices[5]. Similarly, SCLC diodes assume that the electric field across the semiconductor is only influenced by the electronic space charge, but the ionic charge may also contribute[22–24]. Time-of-flight (TOF)[8] is a transient electrical method which determines the photo-induced charge carrier mobility, but again requires the explicit assumption that the electric field is constant across the semiconductor layer. Using a certain range of probe frequency, optical-pump terahertz-probe (OPTP) spectroscopy[9,10,16–18] and time-resolved microwave conductivity (TRMC)[12,13,19,20] evaluate the photo-induced carrier mobility of perovskite thin films using non-contact transient optical probes. OPTP photo-conductivity can determine reliable "early-time" mobility values, under high excitation density regimes. However, since the mobility is estimated from the photo-conductivity peak within the first few tens of ps after excitation, the carriers have not had sufficient time to explore a sizeable volume of the material and the mobility estimated for metal halide perovskites is explicitly short-range, on the order of a few to tens of nm. In contrast, TRMC probes the mobility at longer times, and typically uses a relatively long ns pulsed excitation. Unlike OPTP mobility measurements, this does therefore assess the charge conduction over a longer range. However, the long and broad excitation pulses employed mean that fast, early-time bi-molecular and Auger recombination can occur during the exci-tation pulse at higher excitation densities, and in order to avoid this only relatively low excitation density regimes can be probed reliably[17,25,26].

In ideal semiconductors, the mobility is independent of charge carrier density until carrier-carrier scattering effects become an important factor[27]. In polycrystalline thin-films, charge transport is additionally influenced by scattering/trapping processes[27,28] that can depend upon charge density, both within the grains and at grain boundaries, leading to lower mobilities. Regarding the long-range charge transport process in perovskite polycrystalline thin-films[29,30], it would be very useful to be able to study this over a broad range of charge carrier densities, relevant for different application regimes.

Herein, we introduce a concise and powerful transient, in-plane photo-conductivity ($\sigma_{Photo}$) methodology, which includes an accurate estimation of the internal free-carrier density during and following photoexcitation. We find that the perovskite thin-film processing method has a strong influence on the long-range charge carrier mobility. By measuring both polycrystalline thin films and single crystals with the same technique, we reveal that

significant advances are still feasible, for the polycrystalline films to reach the ultimate mobility values attainable in the single-crystal form. This knowledge and methodology will prove invaluable for optimising active layers for in-plane devices, such as interdigitated back contact solar cells, LEDs and FETs, and for increasing the active layer thickness in sandwich-structured PV cells and radiation detectors. Furthermore, we show that the same correction factors that we have introduced here, which account for both early time recombination and exciton-to-free-carrier branching ratios, are also applicable to TRMC data to accurately estimate the charge carrier mobility over a broad range of charge carrier densities.

## Results

**Transient mobility measurements**. With transient photo-conductivity (TPC), which we depict in Fig. 1a, we can estimate the photo-induced charge carrier mobility within a material at a particular pump modulation frequency and fluence[4,28]. In our study, we photo-excite our devices with an Nd-YAG laser with a 470 nm, 3.74 ns full-width-half-maximum (FWHM) pulse at 10 Hz repetition rate. This output pulse excitation source becomes an optical trigger for the digital oscilloscope and is attenuated in its intensity by neutral density filters and diverged by a lens in front of the sample to uniformly illuminate the perovskite films between in-plane electrodes, with excitation and

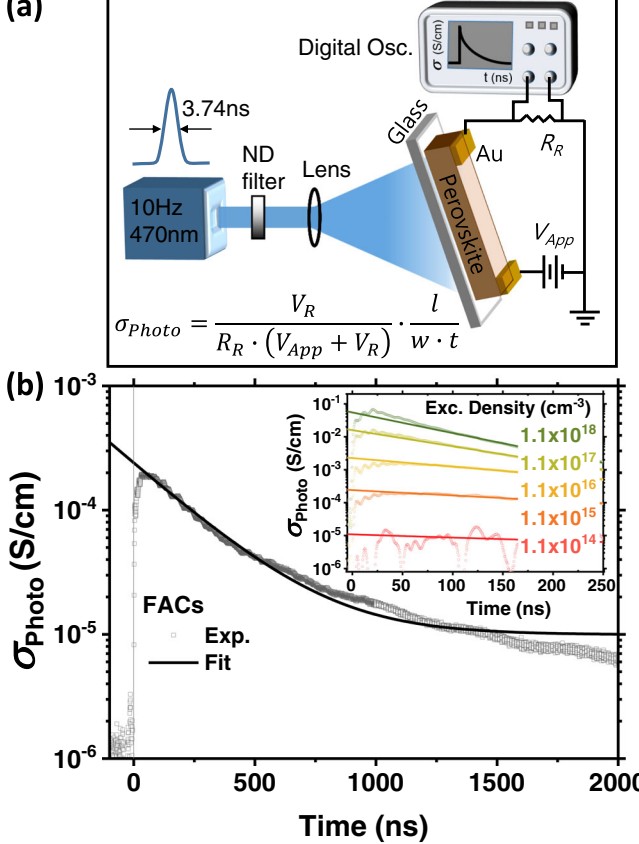

**Fig. 1 Measurement and calculation of transient photo-conductivity (TPC). a** Schematic illustration of the experimental setup and sample structure with in-plane electrodes used in this study. **b** Photo-induced transient change of conductivity and exponential decay fitting for FACs polycrystalline perovskite thin-films. Photo-conductivity ($\sigma_{Photo}$) at $t = 0$ is extrapolated from the fitted plot to calculate mobility. Inset shows $\sigma_{Photo}$ decays from the experiment and fitting with various excitation densities (cm$^{-3}$).

device area of up to a few mm$^2$. For photoconductivity ($\sigma_{Photo}$) measurements, we probe the region between the in-plane electrodes by applying a small DC bias voltage to the electrodes to induce an electric field, and monitor the change in resistance of the in-plane device during and following photoexcitation. We use a digital oscilloscope to detect the current flow by monitoring the transient voltage drop across a small resistor, which is in series with the in-plane conductivity device and voltage source. We then calculate the $\sigma_{Photo}$ using the following relationship[4,31,32],

$$\sigma_{Photo} = \left(\frac{V_R}{R_R\left(V_{App} - V_R\right)}\right)\left(\frac{l}{w \times t}\right), \qquad (1)$$

where $V_R$ is the monitored voltage drop through the fixed resistor, $R_R$, $V_{App}$ is the externally applied bias voltage, $l$ is the distance between the in-plane electrodes, $w$ is the channel width and $t$ is the perovskite film thickness. Although there might be a diffusion in z-direction (towards the opposite side of illumination), the applied electric field between the electrodes results in charge carrier drift in the plane of the film, because the film and in-plane device have a high aspect ratio of ~1000 between the in-plane electrode spacing (400 µm) and film thickness (~400 nm). We demonstrate that transient photo-conductivity decay profiles are independent of illumination direction in Supplementary Fig. 1.

Equation 1 assumes that the electric field is uniform between the in-plane electrodes. Since metal halide perovskites contain mobile ionic species, we need to verify if this assumption is valid. We used a Kelvin probe to measure the surface potential between the in-plane electrodes under an externally applied bias voltage of similar magnitude to that used in our TPC experiments (<5 mV µm$^{-1}$, which is 3 orders of magnitude smaller than that for solar cell characterization), as a function of the distance between the electrodes, with and without excitatiom by continuous-wave light (Supplementary Fig. 2). We ascertain that the potential drops approximately linearly between the electrodes, indicating an approximately uniform electric field across the channel. We do note that under light there is a small drop in potential near the electrodes, but this is <10% of the total potential drop across the channel and will hence only make a small difference to our determined photo-conductivities. This justifies our use of Eq. 1. We note that we measured the $\sigma_{Photo}$ value after 1 min of illumination in air, in order to reach a steady-state condition, following the time-dependent photo-doping, or light soaking effect[4,33].

To demonstrate our TPC technique, we prepared polycrystalline FA$_{0.83}$Cs$_{0.17}$Pb(I$_{0.9}$Br$_{0.1}$)$_3$ (hereon referred to as FACs)[34,35] perovskite films. We spin-coated and cured FACs films on top of glass substrates followed by evaporation of in-plane Au (75 nm) electrodes on top of the perovskite films. The distance between the Au in-plane electrodes, the channel length, was 400 µm or 4 mm. We outline additional information on the device preparation, and also demonstrate the suitability of two-probe (as opposed to 4-wire sense) in-plane electrodes in Supplementary Fig. 3.

We plot the transient $\sigma_{Photo}$ of an FACs perovskite film as a function of time in Fig. 1b. With the pulsed laser excitation, a high charge carrier density is temporarily generated, which decays as a function of time due to internal carrier recombination processes. We note that carriers could also be removed by sweep-out to the electrodes. As a simple estimation of charge carrier transport distance during the measurement time, with mobility ranging from 1 to 10 cm$^2$/Vs and an electric field of 5 mV/µm, we calculate that the drift velocity ($v = \mu E$) ranges from 0.5 to 5 µm/µs. Therefore, during the decay (~1 µs) there is very little charge carrier sweep-out, since the channel is 400 µm wide. Therefore, the

photoconductivity signal is predominantly due to the displacement current in the electronic circuit (analogous to the situation for time-of-flight mobility measurements), and the photoconductivity decays take place because of recombination within the perovskite thin-films (single crystals) and not due to carrier sweep-out.

The measured $\sigma_{Photo}$ is a simple product of mobility and photo-induced charge carrier density[28,32],

$$\sigma_{Photo} = e\left(\mu_n n + \mu_p p\right), \qquad (2)$$

where $e$ is the elementary charge, $n$ and $p$ are densities of photo-generated electrons and holes, respectively, $\mu_n$ and $\mu_p$ are the average mobilities for corresponding charge carrier types. The technique cannot distinguish between electron and hole populations, and therefore provides a measurement of the sum-of-mobilities of electrons and holes, which following convention we term $\Sigma\mu$. Further, we neglect selective trapping of carriers and consider $n \approx p$ during early decay time. Thus, one can express the transient photoconductivity with the simple relation:

$$\sigma_{Photo}(t) = e\phi n(t)\Sigma\mu, \qquad (3)$$

where $\phi$ is the fraction of absorbed photons which generates mobile free carriers.

To extract the mobility from the transient photoconductivity measurements, one therefore requires an accurate estimation of the carrier density as a function of time.

To estimate the number of incident photons absorbed by the films, we measured their absorptance using a spectrophotometer equipped with an integrating sphere. Despite the large absorption coefficients of the perovskites (about $10^4 \sim 10^5$ cm$^{-1}$ at 470 nm), and a film thickness around 400 nm in this study, we found that the fraction of incident photons absorbed does not exceed 75%, due to the strong reflectance associated with the moderately high refractive index of the perovskites (see Supplementary Fig. 4). We assume that each photon absorbed generates an electron-hole pair, of which a fraction $\phi$ corresponds to mobile free carriers.

A common method to estimate the magnitude of the transient conductivity perturbation is to extrapolate the value back to zero time ($t = 0$, immediately after the laser pulse) using an empirical fit, as we show in Fig. 1b. Thus, we can determine the $\sigma_{Photo}$ at $t = 0$. We perform the $\sigma_{Photo}$ measurement using various excitation densities for the same device, and plot $\sigma_{Photo}$ as a function of time for each excitation density, with extrapolated curves, in the inset of Fig. 1b.

With knowledge of the initial excitation density and $\sigma_{Photo}$ at $t = 0$, it is possible to make a first estimation of the mobility, following the relationship (2). Here, we assume that minimal charge carrier recombination occurs during the laser excitation pulse. Therefore, if we know the total charge density $I_{Tot}$ (excitation density, determined from the laser fluence and absorptance of the films), we can estimate the charge carrier mobility for both electrons and holes ($\Sigma\mu$) through the following simple relation[28],

$$\sigma_{Photo(t=0)} = eI_{Tot}\phi\Sigma\mu, \qquad (4)$$

We note that here, the sum-of-mobility $\Sigma\mu$, which is also usually used for microwave and THz mobility estimations, is approximately double the value of the average of electron and hole mobilities.

We show $\phi\Sigma\mu$ versus excitation density in Fig. 2b, which we determine to be in the range 0.04 and 1.4 cm$^2$ V$^{-1}$ s$^{-1}$, over 5 orders of magnitude in excitation density. Our observation is that the $\phi\Sigma\mu$ estimated in this manner for FACs films is significantly dependent upon excitation density, monotonically reducing with increasing excitation density.

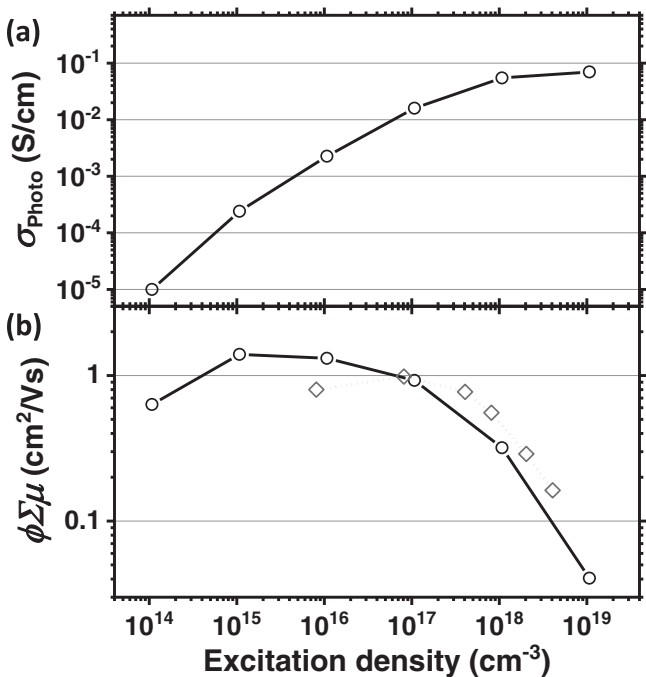

**Fig. 2 Evaluation of excitation density-dependent optoelectronics properties of FACs perovskite thin films. a** $\sigma_{Photo(t=0)}$ calculated from the extrapolation as a function of excitation density (cm$^{-3}$). **b** Charge carrier mobility ($\phi\Sigma\mu$, sum-of-mobilities of electron and hole) determined by TPC and TRMC (for circle and diamond, respectively, lines are only to guide the eyes) as a function of excitation density (cm$^{-3}$). For TPC, the $\sigma_{Photo(t=0)}$ is used for Eq. 4, ($\sigma_{Photo(t=0)} = el_{Tot}\phi\Sigma\mu$). The charge carrier mobilities ($\phi\Sigma\mu$) from both measurements are in a good agreement for the entire range of excitation densities.

We also compare our measurement with TRMC[12,19,20] for similar FACs films, in order to assess if our estimated mobility is in line with non-contacted transient optical probe methods. For the TRMC set up, we employ a ~5 ns FWHM excitation at 532 nm wavelength, and also make an assumption that no early-time recombination occurs during the excitation pulse, as is standard practice for this methodology. We plot mobility data determined from TRMC for the same FACs perovskite films in Fig. 2b (grey diamond) along with our TPC mobility, which closely coincides. We also show the time-resolved decay profile of microwave conductivity data in Supplementary Fig. 5.

**Accounting for early-time recombination and free-carrier fraction**. We now return to our central assumption above; that all photons absorbed from the laser pulse generate electrons and holes, at $t = 0$ and that we can extrapolate $\sigma_{Photo}$ back to $t = 0$ to determine the corresponding mobility. In this assumption, we hypothesize negligible recombination occurring within the time frame of the pulsed laser excitation (3.74 ns FWHM), implying that the excitation density is equal to the total electron density. However, the early-time recombination occurring concurrently with photoexcitation, may also reduce the peak charge carrier population generated during the pulse period[36]. In addition, there will be an equilibrium population of excitons and free-carriers, which will become increasingly significant under high excitation densities. Both effects, early-time recombination and exciton population, will act to reduce the free charge carrier density and hence photo-conductivity, with respect to increasing photo-excitation density (see Supplementary Fig. 6 for an illustration of these effects). They will therefore lead to an underestimation of mobility, with increasing impact at higher excitation densities.

Many of the fundamental optoelectronic properties of metal halide perovskites can be well explained via well-established semiconductor theory, and this includes bi-molecular band-to-band recombination[37] and the balance between excitons and free-carriers in these materials[38]. With knowledge of the bi-molecular and Auger recombination rates for this absorber layer, and assuming the rate equation for charge recombination follows[17],

$$\frac{dn(t)}{dt} = -k_1 n - k_2 n^2 - k_3 n^3, \qquad (5)$$

we can account for early-time recombination. Bi-molecular and Auger rate constants, $k_2$ and $k_3$ respectively, are intrinsic materials properties and are often available from previous reports. For similar FACs films to those that we study here, they have been estimated to be ~$0.2 \times 10^{-10}$ cm$^3$ s$^{-1}$ and $0.2 \times 10^{-28}$ cm$^6$ s$^{-1}$ respectively[39]. $k_1$ is associated with trap-assisted recombination (via the Shockley-Read-Hall mechanism) and depends strongly on the film fabrication process. Therefore, it is important to estimate this recombination rate for the same films used in the photo-conductivity measurements. Using the simple kinetic model of Eq. 5, $k_1$ can be obtained by fitting the photo-conductivity decays (vide infra). Alternatively, it is possible to use time-resolved photoluminescence spectroscopy (TRPL), to measure PL decays for various excitation densities and obtain $k_1$ and $k_2$ for the different films. We show that both methods are in good agreement in the Supplementary Note 1 on the fitting of the recombination constants (Supplementary Figs. 7–9).

Next, we estimate the free-carrier fraction ($\phi$) as a function of total charge carrier density using the Saha equation[38],

$$\frac{\phi^2}{1-\phi} = \frac{1}{n}\left(\frac{2\pi\mu kT}{h^2}\right)^{\frac{3}{2}} e^{-\frac{E_B}{kT}}, \qquad (6)$$

where $n$ is the total excitation density ($n = n_{free-carriers} + n_{excitons}$), $\mu$ is the reduced effective mass[40], $T$ is the temperature, $k$ is Boltzmann's constant, $h$ is Plank's constant, and $E_B$ is the exciton binding energy.

We estimate the exciton binding energy ($E_B$) for the FACs thin-film which we have studied, via fitting the absorption profile to the Elliot model (Supplementary Fig. 10), to be ~11 meV. In Supplementary Fig. 11, we show the free-carrier fraction versus total charge population density at room temperature for a semiconductor with an exciton binding energy of 11 meV. We can now calculate the free-carrier density as a function of time, via multiplying the total carrier density by the corresponding free-carrier fraction, $\phi$, and we plot this in Fig. 3a.

Using the full model to calculate the carrier density, which takes into account carrier recombination and exciton/free-carriers balance, we now find the peak free carrier density, and we show an example for a selected excitation density in Fig. 3a. We can now correct the mobilities estimated via extrapolation to $t = 0$ ($\phi\Sigma\mu(t=0)$) by the peak free carrier fraction to obtain the sum-of-mobilities $\Sigma\mu_{(t=peak)}$. Using the peak free carrier density we can then calculate a corrected photo-conductivity $\sigma_{Photo(t=peak)}$.

In Fig. 3b–c, we report the photo-conductivities and sum-of-mobilities as a function of the maximum carrier density achieved during the pulse ($t = t_{peak}$), and compare with our previous estimation obtained by extrapolation to $t = 0$. We note that the time for reaching the peak of free-carrier density ($t = t_{peak}$) moves to longer times at lower excitation densities, which is simply due to the reduced contribution of the early-time recombination to reducing the carrier density (Supplementary Fig. 12).

From Fig. 3b, we see that the photo-conductivity with carrier density estimated at $t = 0$ increases sub-linearly for free-carrier densities above $10^{16}$ cm$^{-3}$ (black squares). Encouragingly, using

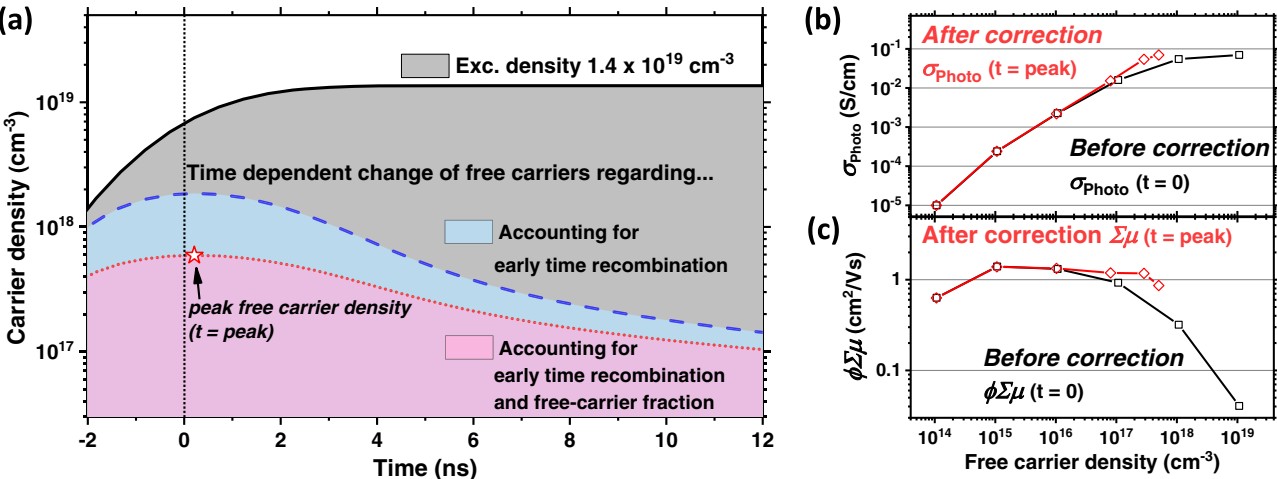

**Fig. 3 Elucidating the effect of free-carrier density on optoelectronics properties of FACs perovskite thin films. a** Photo-induced carrier population changes as a function of time: Assuming all absorbed photons lead to free-carriers (black solid line, also represents Supplementary Fig. 6a) and assuming free-carrier density reduced by early-time recombination (blue dashed line, also represents Supplementary Fig. 6b), and assuming free-carrier density reduced by early time recombination and exciton-to-free-carrier-ratio (red dotted line, also represents Supplementary Fig. 6c). Star symbol is a corresponding time (ns) for peak free-carrier density. **b** Photo-conductivity before ($\sigma_{\text{Photo}(t=0)}$, black square) and after ($\sigma_{\text{Photo}(t=peak)}$, red diamond) correction as a function of free-carrier density. **c** Charge carrier mobility before ($\phi\Sigma\mu_{(t=0)}$, black square) and after ($\Sigma\mu_{(t=peak)}$, red diamond) correction as a function of carrier density, respectively, determined by TPC. All lines (3b–c) are only to guide the eyes.

the corrected carrier density estimated at $t = t_{\text{peak}}$, the photo-conductivity increase now linearly over the 4 orders of magnitude in free-carrier density (red diamonds). Similarly, the estimated $\Sigma\mu$, which was decreasing without correction, is now largely invariant over 4 orders of magnitude in carrier density (Fig. 3c). Notably however, it is important to correct for the exciton to free-carrier branching ratio, otherwise, this invariance of charge carrier mobility with carrier density is not observed.

By applying our free-carrier density estimation methodology to TRMC data, we also show in Supplementary Fig. 13 on the same graph, our re-evaluated $\Sigma\mu$ (dotted pink line) of a similarly prepared FACs film as a function of free-carrier density (along with $\phi\Sigma\mu$ (dotted grey line) vs. excitation density). Consistently, the re-evaluated mobilities of FACs films measured by TPC and TRMC methods are in close agreement, indicating that our accurate carrier density evaluation is powerfully applicable for other time-resolved methods. The coincidence of our sum-of-mobility estimations $\Sigma\mu$, between the two methods also indicates that, particularly at higher carrier density, TRMC suffers from the same overestimation in carrier density, hence the underestimation in charge carrier mobility as shown in Supplementary Fig. 13. We do note that Oga et al[19]. and Labram et al[20]. did interpret their dropping mobility with increasing carrier density to be due to higher order (e.g. bi-molecular and Auger) recombination processes during the ns laser excitation, but did not proceed to correct for this. We also note that we have not previously accounted for this factor in electronic contact transient mobility measurements, which explains our previously observed apparent charge density-dependent carrier mobility, as an artefact of the methodology[28].

**Influence of composition and processing upon charge carrier mobility**. Using TPC, we have estimated above the long-range mobility for mixed cation mixed halide FACs films to be in the range of 1 cm² V⁻¹ s⁻¹ over a wide range of carrier density, with a 400 μm in-plane electrode distance, which is also consistent with the value we obtain using a much wider channel length (4.0 mm) in Supplementary Fig. 3. Incidentally, this demonstrates that the contact resistance at the interface between Au electrode and perovskite is relatively small, in comparison to the channel

resistance, indicating that the two-points probe measurement is suitable for photo-conductivity measurement in this study. Beyond the specific composition of material which we have studied, there are many other processing routes and perovskite compositions used in the research community. Here, we assess if the long-range charge carrier mobility of the perovskite layer is influenced by the thin-film preparation methodology and/or composition. We prepared three MAPbI₃ perovskite films using different fabrication processes, as well as different mixed cation, mixed halide perovskite films, which are representative of state-of-the-art perovskite absorbers used in the research community; MA-based single cation samples: MAPbI₃₋ₓClₓ coated from dimethylformamide (DMF) as solvent (which we term, DMF route)[41], MAPbI₃ from the acetonitrile solvent with the addition of methylamine (termed ACN/MA route)[42], MAPbI₃ from DMF as the solvent employing lead-acetate as the lead source precursor with the addition of hypophosphorous acid (HPA) (termed DMF/HPA route)[43], and mixed cation samples: FA₀.₈₃Cs₀.₁₇Pb(I₀.₉Br₀.₁)₃ (termed, FACs)[34,35] and (FA₀.₈₃MA₀.₁₇)₀.₉₅Cs₀.₀₅Pb(I₀.₉Br₀.₁)₃ (termed, FAMACs)[34,44] which are processed from a DMF/dimethylsulfoxide (DMSO) mixed solvent via the "anti-solvent quenching" method (See the "Methods" section for the respective deposition procedures).

We performed the TPC measurements, and plot the $\sigma_{\text{Photo}}$ of these perovskite films as a function of excitation density in Supplementary Fig. 14. Accounting for both early-time recombination[18,37,39] and free-carrier fraction, in Fig. 4a, we plot $\Sigma\mu$ of these different samples (in addition to the FACs film) versus free-carrier density. All films exhibit a lower mobility as compared to the DMF route film (MAPbI₃₋ₓClₓ), with the lowest $\Sigma\mu$ of the DMF/HPA route film, which has relatively small crystalline grains[43] as compared to the MA-based single cation samples using different fabrication route. For the mixed cation samples with different composition, the FACs samples shows higher $\Sigma\mu$ than the FAMACs film. Our estimated $\Sigma\mu$ of all perovskite films is approximately invariant with free-carrier density, and at average values of 6.7 ± 0.5, 1.1 ± 0.1, 0.3 ± 0.1, 0.6 ± 0.04 and 1.2 ± 0.1 cm² V⁻¹ s⁻¹ for DMF, ACN/MA, DMF/HPA routes, FAMACs and FACs films respectively. Notably, we observe around one order of magnitude difference in long-range

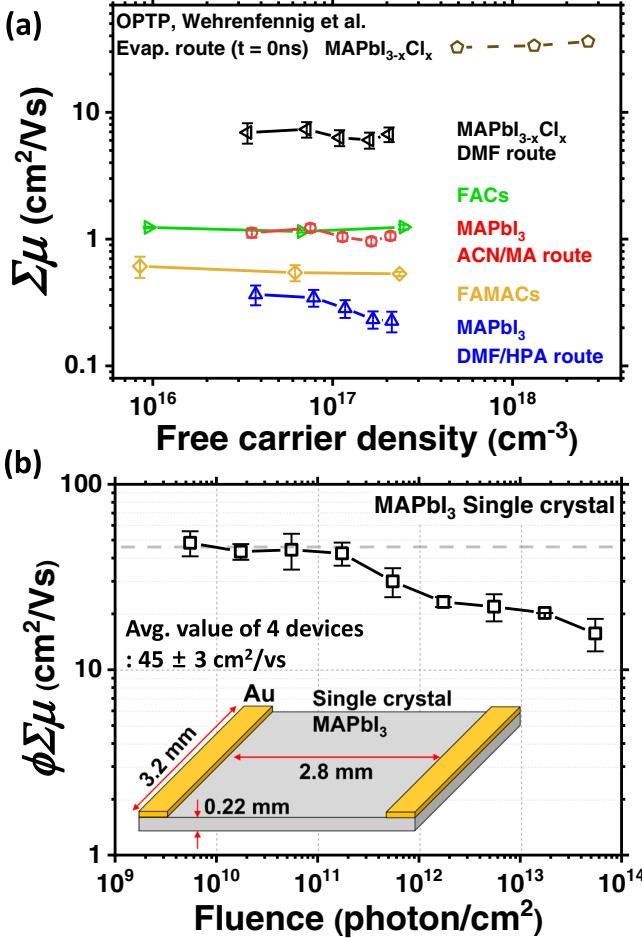

**Fig. 4 Evaluation of carrier density invariant lateral mobility of metal halide perovskite. a** Mobility of metal halide perovskite thin-films prepared via different methodologies as a function of free-carrier density (cm$^{-3}$); $CH_3NH_3PbI_{3-x}Cl_x$ from dimethyl formamide solvent (DMF route, black line), $CH_3NH_3PbI_3$ from the acetonitrile solvent with the addition of methylamine (ACN/MA route, red line), $CH_3NH_3PbI_3$ from DMF as the solvent employing lead-acetate as the lead source precursor with the addition of hypophosphorous acid (HPA) (DMF/HPA route, blue line), $FA_{0.83}Cs_{0.17}Pb(I_{0.9}Br_{0.1})_3$ (FACs, green line) and $(FA_{0.83}MA_{0.17})_{0.95}Cs_{0.05}Pb(I_{0.9}Br_{0.1})_3$ (FAMACs, yellow line). For comparison, we reproduce mobilities of $CH_3NH_3PbI_{3-x}Cl_x$ (evaporation route) determined through non-contact OPTP (brown dashed line). **b** Mobility of $CH_3NH_3PbI_3$ (MAPbI$_3$) thin single crystals as a function of fluence (photon cm$^{-2}$). Average mobility of three devices is $45 \pm 3$ cm$^2$ V$^{-1}$ s$^{-1}$ (grey dashed line). The inset illustration shows device structure. All errors shown are statistical errors from four different devices.

mobility between our MA–single cation samples prepared via different processing routes. These differences may be related to properties such as perovskite composition, grain sizes[1,42,43], crystallinity and crystal orientation, and also may be explained by trapping/detrapping processes[28], grain-boundary scattering[45,46], and charging at grain boundaries. Investigating precisely which material properties most strongly influence the long-range charge carrier mobility is subject to ongoing work.

We believe that these now represent a good estimation of the lateral charge carrier mobility within these perovskite films, and should therefore correlate with the relevant mobility for understanding optoelectronic devices that rely on long-range charge transport, on the range of hundreds of nanometres to hundreds of microns.

A question which regularly puzzles the community, is whether polycrystalline thin-films of metal halide perovskites have properties similar to macroscopic single crystals, or whether single crystals really have superior properties, such as higher charge carrier mobility and lower defect densities. It is not usually possible, however, to compare these different forms of material, via the same methodology. Furthermore, electronic measurements of single crystals, such as space-charge-limited-current, are plagued with uncertainties, unless ionic diffusion and redistribution effects are carefully accounted for[22–24]. To determine how the long-range charge carrier mobility in single crystals compares to polycrystalline thin films, we prepared a MAPbI$_3$ single crystal and estimated the lateral mobility using the TPC measurement methodology[47,48]. In the case of thin films, the extremely high aspect ratio between in-plane electrodes to film thickness, allowed us to assume that the density of charge is approximately uniform in the out-of-plane direction, neglecting the effect of carrier diffusion in the vertical, z-direction. In contrast, for thick single crystals, where photoexcitation occurs predominantly near one surface, in order to accurately correct for free-carrier population following photoexcitation, we need to also account for charge carrier diffusion in the z-direction, which would act to reduce the bi-molecular recombination rate as a function of time. This is presently beyond the scope of this work. However, from our measurements on the thin films, we know that if we go to low enough excitation fluence, we can neglect bi-molecular recombination and the exciton fraction. Therefore, we prepared large area multiple millimetre size MAPbI$_3$ single crystals with hundreds of micrometre thickness, and deposited in-plane electrodes with ~3 mm channel lengths, as we depict in Fig. 4b. We performed the TPC measurement over 4 orders of magnitude of excitation fluence, and plot $\phi\Sigma\mu$ as a function of fluence, estimated from $\sigma_{Photo}$ at $t = 0$, without correction in Fig. 4b. As we revealed in Fig. 3c, the sum-of-mobilities of electron and hole at low excitation fluencies show consistent values before and after correction over a few orders of magnitude in excitation fluence. Knowing the free-carrier fraction ($\phi$) is equivalent to 1 at lower fluence, all charge carriers exist as free carriers. Therefore, below a certain fluence ($10^{11}$ photon cm$^{-2}$) in Fig. 4b, we determine the $\Sigma\mu$ of MAPbI$_3$ single crystal to be $45 \pm 3$ cm$^2$ V$^{-1}$ s$^{-1}$ as an average value from the TPC measurement of three individual devices (See the Supplementary Table 2). This is clear evidence that the long-range charge carrier mobility in single crystals is much higher than in the polycrystalline thin films, and hence significant room for further improvement of the thin films still remains.

As a further indication of how much improvement in charge carrier mobility is feasible for the polycrystalline films, OPTP mobility values, which probe short-range mobility within MAPbI$_3$ films, are in the range of 10–30 cm$^2$ V$^{-1}$ s$^{-1}$[17,18,29], a full two orders of magnitude higher than the long-range mobility we determine here for state-of-the art FACs and FAMACs films. Both our single-crystal measurements and comparison to the OPTP measurements imply that significant improvements should be possible in the present "state-of-the-art" polycrystalline perovskite solar absorbers, which are evidentially "sub-standard" in terms of long-range charge carrier mobility. It is worth noting that the long-range charge carrier mobility will be most relevant in optoelectronic device structures which require carrier transport over distances greater than ~100 nm, and is until now not clearly derived. We now need to focus efforts on understanding parameters which can be tuned to improve this long-range mobility and understand the impact of long-range charge carrier mobility upon the electronic properties of different devices and device configurations. Once we have revealed and gained control over the key factors which limit the long-range mobility, for

instance, charge trapping and scattering at grain boundaries, we should expect long-range mobilities in metal halide perovskites to approach the values of single crystals.

## Discussion

We have presented a concise and powerful transient optoelectronic technique to determine that the long-range mobility within metal halide perovskite films is invariant with changes in carrier density in the range applicable to most optoelectronic devices. We accurately estimate the internal free-carrier density using a simulation of charge population rise and decay accounting for early-time recombination and the free-carrier versus exciton population. Knowing both free-carrier density and the measured $\sigma_{Photo}$, we evaluate the long-range mobility within MAPbI$_3$ films prepared via different processing routes, and FA$_{0.83}$Cs$_{0.17}$Pb(I$_{0.9}$Br$_{0.1}$)$_3$, (FA$_{0.83}$MA$_{0.17}$)$_{0.95}$Cs$_{0.05}$Pb(I$_{0.9}$Br$_{0.1}$)$_3$ as a function of free-carrier density. We determine the mobilities to be in the range from 0.3 to 6.7 cm$^2$ V$^{-1}$ s$^{-1}$ depending upon the composition and thin-film preparation route, which is significantly lower than the long-range sum-of-mobilities for MAPbI$_3$ single crystals, which we determine to be 45 ± 3 cm$^2$ V$^{-1}$ s$^{-1}$. Thereby we suggest that further improvement in the processing of polycrystalline metal halide perovskite thin-films is needed to obtain 'single-crystal'-like charge carrier mobility. We also show that re-evaluated mobility data determined from TRMC coincides almost perfectly with mobilities determined by our TPC method, indicating the applicability of our free-carrier density estimations to a broader set of spectroscopic techniques. The required input parameters for our methodology are the bi-molecular recombination rate and the exciton binding energy, which for a conventional semiconductor including metal halide perovskites can both be determined from absorption spectra[49]. Beyond metal halide perovskites, we expect that our methodology will be especially interesting for estimating the long-range charge carrier mobility in semiconductor materials where the long-range mobility is expected to vary considerably in comparison to the short-range mobility, such as for quantum dots, carbon materials, semiconducting organic molecules and metal oxides.

## Methods

**Preparation of FA$_{0.83}$Cs$_{0.17}$Pb(I$_{0.9}$Br$_{0.1}$)$_3$ and (FA$_{0.83}$MA$_{0.17}$)$_{0.95}$Cs$_{0.05}$Pb(I$_{0.9}$Br$_{0.1}$)$_3$ perovskite films**[4,34,35]. To form the mixed-cation lead mixed anion perovskite precursor solutions, caesium iodide (CsI, Alfa Aesar), formamidinium iodide (FAI, GreatCell Solar), methylammonium iodide (MAI, GreatCell Solar), lead iodide (PbI$_2$, TCI) and lead bromide (PbBr$_2$, Alfa Aesar) were prepared in the way corresponding to the exact stoichiometry for the desired (FA$_{0.83}$MA$_{0.17}$)$_{0.95}$Cs$_{0.05}$Pb(I$_{0.9}$Br$_{0.1}$)$_3$ (termed FAMACs) and FA$_{0.83}$PbCs$_{0.17}$ (I$_{0.9}$Br$_{0.1}$)$_3$ (termed FACs) compositions in a mixed organic solvent of anhydrous N,N-dimethylformamide (DMF, Sigma-Aldrich) and dimethyl sulfoxide (DMSO, Sigma-Aldrich) at the ratio of DMF:DMSO = 4:1. The perovskite precursor concentration used was 1.40 M. The deposition of perovskite layers was carried out using a spin coater in a nitrogen-filled glove box with the following processing parameters: starting at 1000 rpm for 10 s and then 5000 rpm for 35 s. Five seconds before the end of the spinning process, a solvent-quenching method was used by dropping 300 μL anisole onto the perovskite wet films. The film average thicknesses are 495 and 550 nm for FACs and FAMACs, respectively.

**Preparation of CH$_3$NH$_3$PbI$_3$ (ACN/MA) perovskite film**. MAPbI$_3$ (termed ACN route) films were fabricated from the ACN/MA compound solvent following a previously published experimental protocol[4,42]. The film's average thickness is 400 nm.

**Preparation of CH$_3$NH$_3$PbI$_{3-x}$Cl$_x$ (DMF) perovskite film**. MAPbI$_3$(Cl) (termed DMF route) films were fabricated from the solvent following a previously published experimental protocol[41]. The film's average thickness is 405 nm.

**Preparation of CH$_3$NH$_3$PbI$_3$ (DMF/HPA) perovskite film**. MAPbI$_3$ (termed DMF/HPA route) films were fabricated from the compound solvent following a previously published experimental protocol[43]. The film's average thickness is 340 nm.

**Preparation of CH$_3$NH$_3$PbI$_3$ perovskite single crystals**. MAPbI$_3$ single crystals were fabricated from the solvent following a previously published experimental protocol[47,48]. The structural parameters of MAPbI$_3$ single-crystal devices are summarized in Supplementary Table 2.

**UV–Vis absorption**. For the FA$_{0.83}$Cs$_{0.17}$Pb(I$_{0.9}$Br$_{0.1}$)$_3$ thin-films, transmission and reflection spectra were measured with a PerkinElmer 1050+ UV-Vis-NIR spectrophotometer equipped with an integrating sphere accessory. The reflection spectra were corrected with a mirror standard to account for the high specular reflectance of the films. The resulting spectra are shown in Supplementary Fig. 4. For all calculations a reflectance of 25% was used, resulting in an absorptance of 75% < 500 nm.

**Time-resolved photoluminescence decay**. The photoluminescence decay curves were acquired on a FluoTime 300 (PicoQuant GmbH) using a PicoHarp 300 as a time-correlated single-photon counting setup (TCSPC) and a pulsed laser diode with an excitation wavelength of 532 nm (LDH-P-C-510, PicoQuant GmbH). The repetition rate of the diode was 0.5 MHz with fluences tuned between 4 and 400 nJ cm$^{-2}$. The films were illuminated from the perovskite side.

**Photoluminescence quantum yield**. The photoluminescence spectra were acquired with a continuous 532 nm laser and a QE Pro High Performance Spectrometer (OceanInsight). The excitation power density was tuned between 3 and 10$^4$ mW cm$^{-2}$ to allow for the measurement of the power-dependent PLQE shown in Supplementary Fig. 8. The films were illuminated from the perovskite side.

**Transient photo-conductivity**. The Nd:YAG laser (Ekspla NT342A) excitation source tuned to a wavelength of 470 nm and pumped at 10 Hz with 3.74 ns pulses (full-width-half-maximum, FWHM) is used at the range of fluences attenuated by an optical density filter to have various carrier densities, as mentioned in the main text. This pulse light illuminated the entire sample area to uniformly excite the film (illustrated in Fig. 1a). A photodetector (Thorlabs, FDS015) was used to detect the pulse source for the optical trigger for transient measurements using a digital oscilloscope. A small DC bias (<5 mV μm$^{-1}$, which is three orders of magnitude smaller than that for solar cell characterization, ~ 3 V μm$^{-1}$) is applied across the in-plane (lateral) electrodes, while the current was monitored by an oscilloscope[4,31,32]. Here, as the contact resistance between perovskite film and Au electrode is fairly small compared to sample resistance, we employed a two-wire conductivity measurement. A fixed resistor was put in series with the sample in the circuit to be <1% of the sample resistance. We monitored the voltage drop across the variable series resistor through a parallel oscilloscope (1 MΩ input impedance) to determine the potential dropped across the two in-plane Au electrodes on the sample. We note that the measured photo-conductivity value was obtained after 1 min illumination in air to minimize the time-dependent photo-doping (total measurement time for one sample is <5 min). Transient photo-conductivity ($\sigma_{Photo}$) was calculated by the equation, $\sigma_{Photo} = \frac{V_R}{R_R(V_{App} - V_R)} \cdot \frac{l}{w \times t}$, where, $V_R$ is the monitored voltage drop through the fixed resistor, $R_R$ is the applied bias voltage, $V_{App}$ is the applied bias voltage, $l$ is the channel to channel length, $w$ is the channel width, and $t$ is the film thickness.

**Time-resolved microwave conductivity**. A microwave-frequency oscillatory electric signal is generated using a Sivers IMA VO4280X/00 voltage-controlled oscillator (VCO). The signal has an approximate power of 16 dBm and a frequency of roughly 8.5 GHz. The oscillatory signal is incident on an antenna inside a WR90 copper-alloy waveguide. The microwaves emitted from the antenna pass through an isolator and an attenuator before they are incident on a circulator (Microwave Communication Laboratory Inc. CSW-3). The circulator acts as uni-directional device in which signals entering from port 1 exit through port 2 and signals entering from port 2 exit through port 3. The incident microwaves pass through a fixed iris (6.35 mm diameter) into a sample cavity. The cavity supports a TE$_{102}$ mode formed by a short section of WR90 waveguide and an ITO-coated glass window that allows optical access to the sample. The sample (thin films deposited on the quartz substrates) is mounted inside the cavity at a maximum of the electric-field component of the standing microwaves, using a PLA sample holder. Microwaves reflected from the cavity are then incident on port 2 of the circulator, exiting through port 3, directed through an isolator, and directed to a zero-bias Schottky diode detector (Fairview Microwave SMD0218). The detector outputs a voltage linearly proportional to the amplitude of the incident microwaves. The signal is amplified by a Femto HAS-X−1-40 high-speed amplifier (gain = ×100). The amplified signal is detected by a Tektronix TDS 3032 C digital oscilloscope. A Continuum Minilite II pulsed Nd:YAG laser is used to illuminate the sample. The laser pulse has a wavelength of 532 nm, a full-width-half-maxima of ~5 ns and a maximum fluence incident on the sample of ~10$^{15}$ photons cm$^{-2}$ pulse. An external trigger link is employed to trigger the oscilloscope ~50 ns before the laser fires. The area exposed to the incident optical pulse is ~25% of the cross-sectional area of the cavity. Changes in the detector voltage under illumination can then be used to extract the relevant TRMC parameters: $\phi\Sigma\mu$ as described elsewhere[20,50].

## Data availability

The data that support the plots within this paper and other findings of this study are available from the corresponding author upon reasonable request.

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

## Acknowledgements

This project was funded by EPSRC, Engineering and Physical Sciences Research Council grants, EP/M005143/1. B.W. acknowledges funding from the European Commission via

a Marie-Skłodowska-Curie individual fellowship (REA Grant Number 706552-APPEL). J.L. also acknowledges the partial support by KETEP (No. 20203040010320).

## Author contributions

J.L. and M.K.-C. equally contributed to this work. J.L. and H.J.S. conceived the concept, designed the experiments, analysed the data and wrote the manuscript. B.W. and H.J.S. guided and supervised the overall project. Y.H.L. and N.S. fabricated solution-processed perovskite thin-films. J.M.B. contributed with discussion and analysis for conductivity measurement. E.A.D. prepared a large area MAPbI$_3$ single crystals. M.J.H. and J.G.L. conducted TRMC measurement and analysed data. B.W. developed the protocol and performed the carrier density simulations and contributed to the spectroscopic data analysis. All authors discussed the results and reviewed the manuscript.

## Competing interests

H.J.S. is co-founder and CSO of Oxford PV ltd, a company commercializing perovskite PV technology. The remaining authors declare no competing interests.
