## [Peer Review File · Nature Communications]

Long-range charge carrier mobility in metal halide perovskite thin-films and single crystals via transient photo-conductivityReviewers' comments:

Reviewer #1 (Remarks to the Author):

The authors present an interesting new approach to measuring the long-range mobility in semiconductor films, and demonstrate some of the detailed physical mechanisms that must be accounted for at high excitation fluence. The paper is well written, and the conclusions are well supported by their data. The manuscript should be accepted after minor revisions, noted below.

1. Figure S1 appears to have an incorrect x-axis. I presume the units should be ns, not s.
2. Can the authors comment on whether photoconductivity decays primarily because of recombination or carrier sweep-out? I presume the former due to the small fields, but this deserves explicit attention in the manuscript. A more direct comparison of the TRMC and TPC dynamics would be revealing in this regard, as the former measurement takes place at zero applied field.
3. Figure S4 shows a timescale that extends to 1 μ s. I advise the authors to be very cautious with timescales longer than 500 ns using the amplifier in question, Femto HAS-X-1-40, as the short-pass filter on its front end causes significant signal distortion at long times.
4. It isn't quite clear what procedure was used to extract "t=0" values of photoconductivity, though it looks as though a simple fit was made to the decay without accounting for the instrument response function (Figure 1b). The instrument response should be explicitly included in such fitting, and the same is true of the modeling they conducted using equation 4. A detailed accounting of instrument response and how it is included in the fitting/modeling is called for. Numerical convolution of the fit function with an appropriate instrument-response function is suitable for analytical fits.
- 4b. More detail on how the numerical modeling was conducted should be provided.
5. It's somewhat surprising that the TRMC and TPC measurements correlate so well with each other given that resistance at grain boundaries would be expected to influence the latter more than the former (at least in the limit of large grains). A discussion of this issue, and perhaps SEM/XRD data to show both SEM domain size, and crystallite size would be useful.

Reviewer #2 (Remarks to the Author):

The authors have measured the long-range charge carrier mobility in metal halide perovskites by transient photoconductivity and revealed how the early-time recombination and branching ratio of excitons and free-carriers influence the determination of the carrier population and the mobility values. The correction has been used to process the measured data and the TRMC results, and the invariance of the charge carrier mobility with carrier density in three typical hybrid perovskite thin films has been discovered. The motivation is interesting, and the manuscript is well written. To measure the long-range charge carrier mobility accurately is critical for semiconductor and I believe this work paths a way to identify this property. Therefore, I would like to recommend publishing this manuscript after the authors address my following concerns:

1. As addressed in the previous publication (EES 12, 169 2019), the long-range mobility is orders lower than the mobility measured by THz and TRMC because the later methods are measuring short-range transport. In the current study, with similar technology however the mobility agrees well with the TRMC results with or without the correction, Fig. 2 and S8. Would the authors explain this conflict?
2. The other conflict comes from the invariance of the charge carrier mobility with carrier density in a wide range from 10^{14} up to 10^{18} cm⁻³. Why the scattering does not influence the long-range mobility at such a high carrier density?

3. The correction of the early-time recombination and branching ratio of excitons and free-carriers is crucial in this manuscript, but I believe it can be better presented. For example, how those two factors are considered and the weight of them are not included in the main text and SI, which makes the curves in Fig. 3d hard to understand. Additional equations and notes may help, even in the SI.
4. Figure 3 (a)-(c) and the caption are confusing. Do the authors mean "free carriers with ..." instead of "without"?

Reviewer #3 (Remarks to the Author):

The manuscript authorized by Lim et al. reports the measurements of long-range charge carrier mobility in $\text{FA}_{0.83}\text{Cs}_{0.17}\text{Pb}(\text{IO}_{0.9}\text{Br}_{0.1})_3$, $(\text{FA}_{0.83}\text{MA}_{0.17})_{0.95}\text{Cs}_{0.05}\text{Pb}(\text{IO}_{0.9}\text{Br}_{0.1})_3$, $\text{CH}_3\text{NH}_3\text{PbI}_{3-x}\text{Cl}_x$ polycrystalline perovskite films and $\text{CH}_3\text{NH}_3\text{PbI}_3$ single crystal via a time resolved microwave conductivity method. The mobility of polycrystals are in the range of $0.3\text{-}7.4\text{ cm}^2\text{V}^{-1}\text{s}^{-1}$ and the single crystal is $\sim 44.6\text{ cm}^2\text{V}^{-1}\text{s}^{-1}$, and conclude that the process methodology of the polycrystals strongly influences the long-range mobility, and that significant advances are still feasible. It is important to accurately estimate the mobility, either in short-range or in long-term, of semiconductors used in optoelectronic devices, and it is also important to find out the exact role of mobility in optoelectronic devices. However, this manuscript is more like a typical measurement on the mobility using the TPC and TRMC methods. The novelty and profundity are not sufficient. Here are some suggestions.

1. The conclusion is not innovative. It is well known that the single crystals would have higher mobility than polycrystals.
2. The measurement method and data-processing method are not special. It is very similar as the non-contact time resolved THz method. It also can provide the accurate estimation of mobility. A lot of similar publications about the mobility estimation using the non-contact time resolved method have been reported by the Oxford Terahertz Photonics group in Oxford University.
3. The description about the OPTP is not precise. It could not only measure the short-range mobility. It is determined by the pulse length and the range of the delay stage. It could even not only measure the transient (photoexcited carrier) mobility. The steady-state mobility could also be estimated by the time-domain THz spectroscopy.
4. The role of the TRMC is over-evaluated. The long-range measurement is not necessarily related to the microwave, THz might be better as the intraband transition of free carrier is in the regime of THz, the long-range is only related to the long excitation pulses, rather than the probe frequency. On the other hand, the long ns pulse excitation would miss out many interband/intraband scattering/cooling/recombination processes occur at fs-ps time, especially in the polycrystals.
5. Why the measurements of polycrystalline films and single crystals are using different chemical stoichiometries. The control experiments are necessary and required.
6. The glass substrate might not be a proper substrate for microwave experiment.
7. The performance of optoelectronic devices built on metal halide perovskites, eg. solar cells, is not only determined by the mobility. The short-circuit current, open circuit voltage, FF are influenced by different factors. It is not comprehensive to focus the influence on the optoelectronic devices only on mobility.

Reviewer 1

Comments to the Author

The authors present an interesting new approach to measuring the long-range mobility in semiconductor films, and demonstrate some of the detailed physical mechanisms that must be accounted for at high excitation fluence. The paper is well written, and the conclusions are well supported by their data. The manuscript should be accepted after minor revisions, noted below.

Our response:

We thank the reviewer for this positive assessment, and thoughtful further comments on our study. We address the reviewer's specific comments and highlight changes that we have made to the manuscript below

Comments 1-1:

Figure S1 appears to have an incorrect x-axis. I presume the units should be ns, not s.

Our response:

Thank you for spotting this, we have revised the figure.

Comments 1-2:

Can the authors comment on whether photoconductivity decays primarily because of recombination or carrier sweep-out? I presume the former due to the small fields, but this deserves explicit attention in the manuscript. A more direct comparison of the TRMC and TPC dynamics would be revealing in this regard, as the former measurement takes place at zero applied field.

Our response:

As the reviewer mentioned, due to small electric field, which is applied in this method, and the relatively large distance between electrodes, the photoconductivity decays take place primarily because of recombination within the perovskite thin-films (or single crystals). As a simple estimation of charge carrier transport distance during the measurement time, with mobility ranging from 1 to 10 cm²/Vs and an electric field of 5 mV/μm, we estimated that the drift velocity ($v = \mu E$) ranges from 0.5 to 5 μm/μs. Therefore, during the decay (~μs) there is very little charge carrier sweep out, since the channel is 400 μm wide. Importantly, for our mobility estimations, which assess the data over the first few hundred ns, there is negligible carrier extraction.

To clarify this point, we added a note in the manuscript on page 7:

'We note that carriers could also be removed by sweep-out to the electrodes. As a simple estimation of charge carrier transport distance during the measurement time, with mobility ranging from 1 to 10 cm²/Vs and an electric field of 5 mV/μm, we calculate that the drift velocity ($v = \mu E$) ranges from 0.5 to 5 μm/μs. Therefore, during the decay (~ 1 μs) there

is very little charge carrier sweep-out, since the channel is 400 μm wide. Therefore, the photoconductivity signal is predominantly due to the displacement current in the electronic circuit (analogous to the situation for time-of-flight mobility measurements, and the photoconductivity decays take place because of recombination within the perovskite thin-films (single crystals) and not due to carrier sweep-out'

Comments 1-3:

Figure S4 shows a timescale that extends to 1 μs . I advise the authors to be very cautious with timescales longer than 500 ns using the amplifier in question, Femto HAS-X-1-40, as the short-pass filter on its front end causes significant signal distortion at long times.

Our response:

We thank the reviewer for pointing on the considerable signal distortion at long times with the amplifier. As far as our paper is concerned, we are only using the earlier time data from the TRMC experiments, for estimation of the carrier mobility. Errors at longer time are not really relevant to this work, although we concur that they could influence estimated decay lifetimes if we were using the TRMC to estimate these (which we are not).

Comments 1-4:

It isn't quite clear what procedure was used to extract " $t=0$ " values of photoconductivity, though it looks as though a simple fit was made to the decay without accounting for the instrument response function (Figure 1b). The instrument response should be explicitly included in such fitting, and the same is true of the modeling they conducted using equation 4. A detailed accounting of instrument response and how it is included in the fitting/modeling is called for. Numerical convolution of the fit function with an appropriate instrument-response function is suitable for analytical fits. 4b. More detail on how the numerical modeling was conducted should be provided.

Our response:

We very much thank the referee for this comment. Since we were "back-extrapolating" the longer time (hundreds of ns) monotonic decay of the photoconductivity trace back to early time, to obtain the $t=0$ photoconductivity values, we assumed that we could neglect the instrument response, which would be over a few ns time-scale, as shown below by our measurement of the ns laser pulse using a silicon diode in the same measurement set up.

However, in considering this carefully, we realised that in performing our procedure, we had not accounted properly for the early time recombination that occurs over the first few tens of ns following excitation. We have therefore quite considerably revised our method for determine the mobility from the transient photoconductivity traces. Instead of calculating the mobility from our extrapolated $t=0$ conductivity and our estimated peak carrier density, we now simulate the entire transient photoconductivity trace, via determining all the carrier recombination constants and exciton binding energy, and use charge carrier mobility as the fitting parameter to match the data. We have revised the manuscript accordingly (presented additional theory, remeasured a lot of the data and refitted the traces), and have also added additional supporting experiments in order to accurately determine the experimental parameters require for the simulations i.e. UV-Vis (Fig. S4), time-resolved PL (TRPL) (Fig. S7), PLQE (Fig. S8-S9). The changes are all highlighted in the revised manuscript.

Comments 1-5:

It's somewhat surprising that the TRMC and TPC measurements correlate so well with each other given that resistance at grain boundaries would be expected to influence the latter more than the former (at least in the limit of large grains). A discussion of this issue, and perhaps SEM/XRD data to show both SEM domain size, and crystallite size would be useful.

Our response:

Prior to making this comparison, we also assumed that the TRMC would determine higher mobilities than our TPC method, precisely due to our assumption that the latter would be more sensitive to long-range diffusion. However, our comparative data of measurements done on similar materials demonstrates that they do estimate similar mobilities. In hindsight, both methods estimate the conductivity over a few to ten to hundreds of ns time range. For TPC, we can estimate that, with a mobility of $1 \text{ cm}^2/\text{Vs}$, and an electric field of $5 \text{ mV}/\mu\text{m}$, the carriers will travel a distance, $d = \mu E t$, of 50 nm within 100 ns .

We prepared the samples in this study with the same protocols as reported in our previous work. Please refer the SEM image in Supplementary Information [Energy Environ. Sci., 2019,12, 169-176]. For the convenience of the reviewer, here we summarize the sizes of individual domains ranging, between 500 nm and 700 nm for $\text{CH}_3\text{NH}_3\text{PbI}_3$ from the

acetonitrile solvent with the addition of methylamine (ACN/MA route), between 236 nm and 585 nm for $\text{FA}_{0.83}\text{Cs}_{0.17}\text{Pb}(\text{I}_{0.9}\text{Br}_{0.1})_3$ (FACs) and between 183 nm and 442 nm for $(\text{FA}_{0.83}\text{MA}_{0.17})_{0.95}\text{Cs}_{0.05}\text{Pb}(\text{I}_{0.9}\text{Br}_{0.1})_3$ (FAMACs).

Reviewer 2

Comments to the Author

The authors have measured the long-range charge carrier mobility in metal halide perovskites by transient photoconductivity and revealed how the early-time recombination and branching ratio of excitons and free-carriers influence the determination of the carrier population and the mobility values. The correction has been used to process the measured data and the TRMC results, and the invariance of the charge carrier mobility with carrier density in three typical hybrid perovskite thin films has been discovered. The motivation is interesting, and the manuscript is well written. To measure the long-range charge carrier mobility accurately is critical for semiconductor and I believe this work paths a way to identify this property. Therefore, I would like to recommend publishing this manuscript after the authors address my following concerns.

Our response:

We thank the reviewer for this positive assessment, and thoughtful further comments on our study. We address the reviewer's specific comments and highlight changes that we have made to the manuscript below

Comments 2-1:

As addressed in the previous publication (EES, 12, 169, 2019), the long-range mobility is orders lower than the mobility measured by THz and TRMC because the later methods are measuring short-range transport. In the current study, with similar technology however the mobility agrees well with the TRMC results with or without the correction, Fig. 2 and S8. Would the authors explain this conflict?

Our response:

As reviewer pointed out, we concluded that TRMC was a short-range method in the previous paper since the results did not agree with our results obtained by Photo-Induced Transmission and Reflection (PITR) method. However, we note that the previous discussion was based on published TRMC results obtained with different samples. Here we performed TRMC with the samples prepared by the sample fabrication route, same recipe and in the same lab, and surprisingly found that the results agreed with our TPC method. In this study, by understanding the recombination decay of photocurrent in TPC method and directly comparing with TRMC, we concluded that TRMC must probe a larger range than we previously assumed. We also note however, than our TPC probes a shorter range than we would intuitively assume. As we described in our response to Reviewer 1, for TPC, we can estimate that with a mobility of $1 \text{ cm}^2/\text{Vs}$, and an electric field of $5 \text{ mV}/\mu\text{m}$, the carriers will travel a distance, $d = \mu E/t$, of 50 nm within 100ns. The majority of our TPC

signal decays over the first few hundred ns, hence we estimating mobility on the tens to hundreds of nm length scale. Both TRMC and our TPC using multiple ns pulse lasers (7ns for the TRMC used here), and estimate the mobility from conductivity traces resolve on the tens of ns timescale. In contrast, optical pump THz mobility is estimated from the peak conductivity estimated only a few tens of picoseconds after excitation. It is not straight forward to estimate the relevant average electric field the charge carriers will be accelerated in during both the THz and microwave pulses, however if these are similar, it is very likely that the distances over which carriers drift in within the timeframe of the THz measurement are much shorter than the distances drifted within the microwave measurement. We also note that THz measurement, which has been performed previously with samples prepared in our laboratory [1, 2], clearly show a much higher mobility, which indicates that this technique is more likely to be shorter-range.

[1] Herz, L. M., Adv. Mater. 26, 1584 (2014).

[2] Herz, L. M., Adv. Funct. Mater. 25, 6218 (2015)

Comments 2-2:

The other conflict comes from the invariance of the charge carrier mobility with carrier density in a wide range from 10^{14} up to 10^{18} cm^{-3} . Why the scattering does not influence the long-range mobility at such a high carrier density?

Our response:

Strong carrier-carrier scattering can limit the carrier mobility. The origin for the faster carrier-carrier scattering in hybrid perovskites is likely to be due to a weaker Coulomb screening compared to GaAs.[1-3] According to the literature, we can expect such a strong carrier-carrier scattering at a charge carrier density higher than 2×10^{19} cm^{-3} . [1-3] Once again, in a fs (ultrafast) pulsed measurement, an absorbed excitation photon fluence of 10^{19} cm^{-3} would lead to the instantaneous generation of that charge density (free carriers and excitons). However, when we account for early-time recombination occurring within the multiple ns timeframe of our excitation pulse, even with a peak absorbed excitation density of 1.4×10^{19} cm^{-3} , we only reach a peak free-carrier density of 6×10^{17} cm^{-3} . Hence, we still remain more than 1 order of magnitude lower than the density at which we would expect to observe significant carrier-carrier scattering events.

[1] Richter, J. M., Nat. Commun. 8, 376 (2017)

[2] Oum, K., Phys. Chem. Chem. Phys. 17, 19238 (2015)

[3] Muller, A., J. Phys. Chem. C 118, 6454, (2014)

Comments 2-3:

The correction of the early-time recombination and branching ratio of excitons and free-carriers is crucial in this manuscript, but I believe it can be better presented. For example, how those two factors are considered and the weight of them are not included in the main text and SI, which makes the curves in Fig. 3d hard to understand. Additional equations and notes may help, even in the SI.

Our response:

Thank you for the suggestion to improve the explanation of the crucial point of this manuscript. As we have responded to Referee 1, upon careful reflection we realised that we hadn't accounted for the early time evolution of excitons and free carriers as accurately as we could have done. We have now considerably revised the explanation based on the extra experiments i.e. UV-Vis (Fig. S4), TRPL (Fig. S7), PLQE (Fig. S8-S9), and calculation as written in the main text (most of the part in 'Accounting for early-time recombination and free-carrier fraction') and SI (Notes on the fitting of the recombination constants'). And we applied the new simulation and calculation process to all data to obtain the accurate values.

Comments 2-4:

Figure 3 (a)-(c) and the caption are confusing. Do the authors mean "free carriers with ..." instead of "without"

Our response:

Thank you for spotting this, we have revised the figure and the corresponding caption. And moved the illustration to the SI.

From previous Figure 3:

Fig. 3. Elucidating the effect of free-carrier density on optoelectronics properties of FACs perovskite thin-films. (a-c) Schematic illustration of photo-induced carrier population change within perovskite films with in-plane electrodes, (a) free-carriers, (b) free-carriers without early-time recombination, (c) free-carriers without excitons and early-time recombination. (d) Photo-induced carrier population changes with various excitation density as a function of time with and without early-time recombination processes. Excitation density (cm⁻³) and free-carrier densities (cm⁻³) with early-time recombination for solid and dash line, respectively, and free-carrier density (cm⁻³) corrected with both early-time recombination and free carrier fraction for dotted line, and corresponding time (ns, star symbol) for peak free-carrier density.

To revised Figure 3:

Fig. 3. Elucidating the effect of free-carrier density on optoelectronics properties of FACs perovskite thin-films. (a) Photo-induced carrier population changes as a function of time. Assuming all absorbed photons lead to free-carriers (black solid line, also represents Fig. S6a) and assuming free-carrier density reduced by early-time recombination (blue dashed line, also represents Fig. S6b), and assuming free-carrier density reduced by early time recombination and exciton-to-free-carrier-ratio (red dotted line, also represents Fig. S6c). Star symbol is a corresponding time (ns) for peak free-carrier density. (b) Photo-conductivity before ($\sigma_{\text{Photo}(t=0)}$, black square) and after ($\sigma_{\text{Photo}(t=\text{peak})}$, red diamond) correction as a function of free-carrier density. (c) Charge carrier mobility before ($\phi\Sigma\mu(t=0)$, black square) and after ($\Sigma\mu(t=\text{peak})$, red diamond) correction as a function of carrier density, respectively, determined by TPC. All lines (3b-c) are only to guide the eyes.

And Figure S6 in SI.

Fig. S6. Schematic illustration of photo-induced carrier population changes within perovskite films with in-plane electrodes, (a) assuming all absorbed photons lead to free-carriers, (b) assuming free-carrier density reduced by early-time recombination, (c) assuming free-carrier density reduced by early time recombination and exciton-to-free-carrier-ratio.

Reviewer 3

Comments to the Author

The manuscript authorized by Lim et al. reports the measurements of long-range charge carrier mobility in $\text{FA}_{0.83}\text{Cs}_{0.17}\text{Pb}(\text{I}_{0.9}\text{Br}_{0.1})_3$, $(\text{FA}_{0.83}\text{MA}_{0.17})_{0.95}\text{Cs}_{0.05}\text{Pb}(\text{I}_{0.9}\text{Br}_{0.1})_3$, $\text{CH}_3\text{NH}_3\text{PbI}_{3-x}\text{Cl}_x$ polycrystalline perovskite films and $\text{CH}_3\text{NH}_3\text{PbI}_3$ single crystal via a time resolved microwave

conductivity method. The mobility of polycrystals are in the range of 0.3-7.4 $\text{cm}^2\text{V}^{-1}\text{s}^{-1}$ and the single crystal is $\sim 44.6 \text{ cm}^2\text{V}^{-1}\text{s}^{-1}$, and conclude that the process methodology of the polycrystals strongly influences the long-range mobility, and that significant advances are still feasible. It is important to accurately estimate the mobility, either in short-range or in long-term, of semiconductors used in optoelectronic devices, and it is also important to find out the exact role of mobility in optoelectronic devices. However, this manuscript is more like a typical measurement on the mobility using the TPC and TRMC methods. The novelty and profundity are not sufficient. Here are some suggestions.

Our response:

We thank the reviewer for the thoughtful discussion on our study. However, we do refute the “novelty and profundity” concerns. Without accurate methods to estimate physical properties of new semiconductor materials, progress can be significantly hindered. Perovskite especially are quite unique in their peculiarities relate to their ionic and electronic properties, and almost all traditional methods to estimate the longer-range mobility in these materials (i.e. space charge limited, time of flight, transistor operation) are plagued with ambiguities. Our novel and simple approach here, for which we also introduce for the first time the need to accurately consider exciton- to-free-carrier-fraction, will significantly assist in the continued progress of metal halide perovskites optoelectronics. We include a point-by-point response to the reviewer’s comments below.

Comments 3-1:

The conclusion is not innovative. It is well known that the single crystals would have higher mobility than polycrystals

Our response:

As the reviewer mentioned, it may be expected that single crystals would have higher mobility than polycrystalline thin films, however, when measured by some methodologies, such as THz photoconductivity, perovskite single crystals often have the same mobility as polycrystalline thin films. Nevertheless in our study, our main message is not to prove that single crystals have a higher mobility, but to measure the long-range mobility, which has not been possible before introducing our TPC method. Indeed, we suggest by TPC method, how much we can improve the mobility values of thin films to approach to the mobility values of single crystal.

Comments 3-2:

The measurement method and data-processing method are not special. It is very similar as the non-contact time resolved THz method. It also can provide the accurate estimation of mobility. A lot of similar publications about the mobility estimation using the non-contact time resolved method have been reported by the Oxford Terahertz Photonics group in Oxford University.

Our response:

We agree with reviewer that there are many reports using ‘non-contact’ time resolved method, which conducts the measurement for thin films deposited on the ‘quartz’ substrate and at the high excitation density, and represents short-range charge transport. Notably, for THz mobility measurements, the optical pulse width is of tens of femto seconds, and the assumption ubiquitously employed is that all absorbed photons contribute to the photoconductivity, the mobility estimated is Φ times the sum of mobilities (for electrons and holes) and Φ is the branching ration of absorbed photons to free carriers. For perovskites Φ is usually assumed to be 1, and there is no accounting for early time recombination (justified by the fact that the measurement takes place over such a short time-scale. In contrast, here we present very simple and powerful method by measuring the samples prepared in glass substrate at broad range of excitation densities which represents the long-range charge transport, and also includes inter-grain mobility. Furthermore, by employing our novel data processing method, we can accommodate the longer pulse width of the ns excitation, equally accounting for free-carrier fraction and early time recombination. Therefore, we believe that TPC method is an innovative way to understand the long-range mobility over device relevant length-scales, also TPC and THz methods will be a good combination in this research community.

Comments 3-3:

The description about the OPTP is not precise. It could not only measure the short-range mobility. It is determined by the pulse length and the range of the delay stage. It could even not only measure the transient (photoexcited carrier) mobility. The steady-state mobility could also be estimated by the time-domain THz spectroscopy.

Our response:

We agree with the reviewer, here we emphasized how the long-range mobility can be comparable to short-range mobility. Therefore, we re-produced the short-range mobility from the literature. We are not disputing that there are further methods to analyse THz transmission spectra, although these are not as common in the field and also carry additional assumptions. Not least of all, to estimate the steady-state carrier density, which is required to convert conductivity into mobility, you need to know all the rates for generation and recombination accurately, under a given irradiance. For sake of not confusing the reader, we only discuss and compare to time resolved THz photoconductivity in this manuscript.

Comments 3-4:

The role of the TRMC is over-evaluated. The long-range measurement is not necessarily related to the microwave, THz might be better as the intraband transition of free carrier is in the regime of THz, the long-range is only related to the long excitation pulse, rather than the probe frequency. On the other hand, the long ns pulse excitation would miss out many interband/intraband scattering/cooling/recombination processes occur at fs-ps time, especially in the polycrystals

Our response:

We can tell that the reviewer is very supportive the THz spectroscopy. We fully agree with the statements above, including that the “longer range” mobility probed by TRMC is due to the ns pulse width. Nevertheless, this (TRMC) is a very similar excitation set up to our methodology, and is used often as a probe of mobility in the field. As of yet, as far as we are aware, no one has accounted for the early time recombination, nor free carrier fraction for TRMC, and hence our comparison here is critical to the ongoing for with TRMC in the research community. Both these methods are complimentary to ultrafast THz spectroscopy, but cheaper to set up and easier for more researchers to follow.

Comments 3-5:

Why the measurements of polycrystalline films and single crystals are using different chemical stoichiometries. The control experiments are necessary and required.

Our response:

We have investigated several perovskite compositions. Also, in order to compare samples with the same stoichiometry, we specifically selected MAPbI₃ which we prepared as single crystal and polycrystalline films using 3 different protocols. Therefore, we believe that the control experiments are already included in our work.

Comments 3-6:

The glass substrate might not be a proper substrate for microwave experiment.

Our response:

We agree with the referee and we note that for the TRMC measurement, we deposited the thin films on the quartz substrates. To clarify this point, we added a note in the manuscript on page 20:

‘The sample (thin films deposited on the quartz substrates) is mounted inside the cavity at a maximum of the electric-field component of the standing microwaves, ...’

Comments 3-7:

The performance of optoelectronic devices built on metal halide perovskites, e.g. solar cells, is not only determined by the mobility. The short-circuit current, open circuit voltage, FF are influenced by different factors. It is not comprehensively to focus the influence on the optoelectronic devices only on mobility.

Our response:

We agree with reviewer’s opinion on device operating parameters. We believe that the performance of optoelectronic devices should be characterized with various parameters as reviewer mentioned. Indeed, in the full device architecture various materials consist of interfaces, which can further complicate charge transport throughout a device. However,

this is a slightly nonsensical comment since here, we propose a new tool to study one of the important parameters to characterise for the understanding and optimisation of perovskite optoelectronic devices. We are by no means trying to suggest that this is the only parameter which is important.

REVIEWER COMMENTS

Reviewer #1 (Remarks to the Author):

I am satisfied with the author's responses to my original suggestions, and recommend publication without further changes.

Reviewer #2 (Remarks to the Author):

To measure the long-range charge carrier mobility accurately is critical for perovskite and other semiconductor materials. However, long-range charge carrier mobility measurement is more complicated and harder for perovskite materials due to their ionic nature. In this study, the authors present a novel and simple method of transient photoconductivity (TPC) to measure the long-range mobility in perovskite materials and revealed the influence of early-time recombination and branching ratio of excitons and free-carriers on the determination of the carrier population and the mobility values, which is reasonable but barely discussed in previous studies. With the correction of the exciton to free-carrier branching ratio used to process the measured data and the TRMC results, the author revealed the invariance of the charge carrier mobility with carrier density in three typical hybrid perovskite thin films.

The author addressed all the reviewer's comments with factual data in this revised manuscript and improved the quality of this work. The long-range mobility of semiconductor materials is an essential parameter for further evaluation of material physical properties, and its accurate measurement is of the great reference value. The motivation is interesting, and the manuscript is well written. I recommend publishing the revised version.

Reviewer #3 (Remarks to the Author):

The TMC method does have some advantages considering that it can eliminate the effect of recombination process. But the novelty is still not sufficient.

As a method to obtain the mobility of perovskite, the discussion in this paper is indeed reasoned, but as a long-range measurement method, the estimation accuracy of the mobility could be lower than that of single crystal perovskite because of effects other than recombination. Additional scattering of carriers between grains results in a further decrease in the mobility of thin film perovskite.

In Fig. S7, the decay trace of TRPL is determined by the recombination rate, while the decay of conductivity is not only affected by the recombination rate, but also includes the fraction of carriers and excitons. Why the lifetime of TRPL is short than that of conductivity under the approximate excitation fluence?

Reviewer 1

Comments to the Author

I am satisfied with the author's responses to my original suggestions, and recommend publication without further changes.

Our response:

We thank the reviewer for the positive assessment and recommendation of our study to be published without further changes.

Reviewer 2

Comments to the Author

To measure the long-range charge carrier mobility accurately is critical for perovskite and other semiconductor materials. However, long-range charge carrier mobility measurement is more complicit and harder for perovskite materials due to their ionic nature. In this study, the authors present a novel and simple method of transient photoconductivity (TPC) to measure the long-range mobility in perovskite materials and revealed the influence of early-time recombination and branching ratio of excitons and free-carriers on the determination of the carrier population and the mobility values, which is reasonable but barely discussed in previous studies. With the correction of the exciton to free-carrier branching ratio used to process the measured data and the TRMC results, the author revealed the invariance of the charge carrier mobility with carrier density in three typical hybrid perovskite thin films.

The author addressed all the reviewer's comments with factual data in this revised manuscript and improved the quality of this work. The long-range mobility of semiconductor materials is an essential parameter for further evaluation of material physical properties, and its accurate measurement is of the great reference value. The motivation is interesting, and the manuscript is well written. I recommend publishing the revised version.

Our response:

We thank the reviewer for the positive assessment and recommendation of our study to be published with revised version.

Reviewer 3

Comments to the Author

The TMC method does have some advantages considering that it can eliminate the effect of recombination process. But the novelty is still not sufficient.

Our response:

We thank the reviewer for the thoughtful comment on our study. However, we do refute the "novelty" concerns. Without accurate methods to estimate physical properties of new semiconductor materials, progress can be significantly hindered. Perovskite especially are

quite unique in their peculiarities related to their ionic and electronic properties, and almost all traditional methods to estimate the longer-range mobility in these materials (i.e. space charge limited current, time of flight, transistor operation) are plagued with ambiguities. Our novel and simple approach here, for which we also introduce for the first time the need to accurately consider exciton-to-free-carrier-fraction, will significantly assist in the continued progress of metal halide perovskites optoelectronics. We note that our positive opinion is also shared with Reviewer 1 and 2.

As stated correctly, the TPC method can be used to estimate long-range mobility from the photoconductivity decays via equation (2). In addition, we show a post-processing of the data taking recombination and exciton-to-free-carrier fraction into account. The resulting mobility should then be widely independent of the charge carrier density, which has not been reported before. We show that this data-processing is applicable to both TPC and TRMC measurements (Fig. S13) indicating that these corrections are generally valid. This is a key novelty of our study and will help researchers using either technique to obtain more reproducible mobility estimations.

Comments 3-1:

As a method to obtain the mobility of perovskite, the discussion in this paper is indeed reasoned, but as a long-range measurement method, the estimation accuracy of the mobility could be lower than that of single crystal perovskite because of effects other than recombination. Additional scattering of carriers between grains results in a further decrease in the mobility of thin film perovskite.

Our response:

We thank the reviewer for pointing out one of the key discussions on our study. We fully agree with the reviewers comment and as mentioned by the reviewer, the long-range mobilities estimated for the polycrystalline perovskite thin-films will be influenced by other factors than recombination alone. As we stated in the main text (lines 339-342): “These differences may be related to properties such as perovskite composition, grain sizes, crystallinity and crystal orientation, and also may be explained by trapping/detrapping processes, grain boundary scattering, and charging at grain boundaries”. Our main message is not to prove that single crystals have a higher mobility, but to measure the long-range mobility in a reproducible and reliable manner, which has not been possible before introducing our TPC method. Indeed, we suggest by TPC method, how much we can improve the mobility values of thin films to approach to the mobility values of the single crystal. Nevertheless, in our study, the good agreement between the mobility obtained for single crystal MAPI via the TPC measurement and thin-film MAPI via the OPTP measurement, indeed gives us high confidence in our interpretations. We therefore particularly noted in line 380 that “This is clear evidence that the long-range charge carrier mobility in single crystals is much higher than in the polycrystalline thin-films, and hence significant room for further improvement of the thin-films still remains”. We are hence positively suggesting that a key finding of our work is that we have quantitatively revealed the difference in long-range mobility between single crystals and polycrystalline thin films, which now needs to be used as a metric to improve and optimise the polycrystalline thin films, and understand factors which limit the long-range mobility. (see lines 362-366). We have slightly adapted the end of the paragraph between lines 387 and 391, to emphasise

the importance of our new measurement method upon enabling future advancements in long range mobility.

Comments 3-2:

In Fig. S7, the decay trace of TRPL is determined by the recombination rate, while the decay of conductivity is not only affected by the recombination rate, but also includes the fraction of carriers and excitons. Why the lifetime of TRPL is short than that of conductivity under the approximate excitation fluence?

Our response:

There are key differences between PL decay measurements and TPC measurements, which are related to the PL intensity scaling with the square of the carrier density (n^2) and the photoconductivity scaling linearly with carrier density. The scaling of PL signal with n^2 has been repeatedly observed for metal halide perovskites and is well understood.[see Nature Communications 2016, 7, 13941 and Nature Photonics 2020, 14, 123–128 for instance]. For our system here, this results in a divergence of the decay rates between the TRPL and TPC measurements as the excitation intensity (and corresponding carrier density) is increased. Two factors contribute to this: 1. The increased fraction of radiative decay (PL) at higher intensity, 2: the diffusion of carriers into the film following excitation (which reduces the local “n” but is not a consequence of recombination). We have added the following note, where we discuss Figure S7 in the SI:

“We note that the faster decay of the TRPL traces at higher charge carrier densities, in comparison to the TPC decay traces, is expected. It is well understood that PL intensity scales with the carrier density squared, when in the bimolecular regime, whereas conductivity will scale linearly with carrier density. Hence, as the carrier density decays after photoexcitation the PL intensity will decay as a square function of the decaying carrier density, until the monomolecular regime is met. In contrast the TPC will decay as a linear function of the decaying carrier density. Furthermore, since the excitation source for the TRPL was 405nm wavelength, this will be predominantly absorbed near the thin film surface. The rapid diffusion of carriers away from the surface will also contribute to the early time decay of the TRPL trace. However, this diffusion is not a recombination process, and will hence not result in a drop in the TPC signal. We finally note that we employed a 470nm photoexcitation for the TPC excitation, which also has a deeper penetration depth. Additionally, during the first < 200 ns TRPL decay (fast radiative recombination as shown in Fig. S7), our home-built TPC setup has an instrument-limited response. It highlights the strength of our post-processing on the data to accurately evaluate the mobility of the mobile charge carriers.”

REVIEWERS' COMMENTS

Reviewer #3 (Remarks to the Author):

The authors demonstrated that the role of excitons cannot be ignored in carrier transport despite the small exciton binding energy in perovskite films, which is innovative and meaningful in improving the accuracy of the measurement. But the long diffusion length of the carriers in perovskite films (Science 342.6156 (2013): 341-344, Science 347.6221 (2015): 519-522) over the grain sizes, the polycrystalline property of the perovskite films will limit the carrier transport. Which means the significance of the long-range mobility measurement in perovskite films will be subdued. It is more suitable to measure the mobility in perovskite single crystal. So I think the authors should take this into account. After that, the article will be recommended publication.

Reviewer 3

Comments to the Author

"The authors demonstrated that the role of excitons cannot be ignored in carrier transport despite the small exciton binding energy in perovskite films, which is innovative and meaningful in improving the accuracy of the measurement. But the long diffusion length of the carriers in perovskite films (Science 342.6156 (2013): 341-344, Science 347.6221 (2015): 519-522) over the grain sizes, the polycrystalline property of the perovskite films will limit the carrier transport. Which means the significance of the long-range mobility measurement in perovskite films will be subdued. It is more suitable to measure the mobility in perovskite single crystal. So I think the authors should take this into account. After that, the article will be recommended publication."

Our response:

We thank the reviewer for their final consideration of our manuscript and for acknowledging the importance of our inclusion of excitonic effects when assessing the mobility in metal halide perovskites. We agree with their comments upon long range mobility in polycrystalline films being limited, or at least influenced by the polycrystalline property, but we disagree with this implying that the "importance of the long-range mobility in perovskites is subdued". In fact, that is a key and important finding and clarification in our manuscript here. In many optoelectronic devices, which are made with polycrystalline films, it is the long-range mobility and diffusion length that will be most relevant, and until now not clearly defined or derived. We do also show and compare the long-range mobility in single crystals (Fig 4) to our polycrystalline films, to precisely highlight this fact. We have reproofed the manuscript and made a few minor changes, which are highlighted in yellow, to ensure that these points are clear to the reader.

We hope that you are now satisfied with our revision and we look forward to hearing your response.